# A Reliable Production System of Large Quantities of [^13^N]Ammonia for Multiple Human Injections

**DOI:** 10.3390/molecules28114517

**Published:** 2023-06-02

**Authors:** Luis Michel Alonso Martinez, Nabil Naim, Alejandro Hernandez Saiz, José-Mathieu Simard, Mehdi Boudjemeline, Daniel Juneau, Jean N. DaSilva

**Affiliations:** 1Radiochemistry and Cyclotron Platform, Centre de Recherche du Centre Hospitalier de l’Université de Montréal (CRCHUM), 900 Rue Saint Denis, Montréal, QC H2X 0A9, Canada; 2Radiopharmaceutical Science Laboratory, CHU de Québec, 2250 Boul. Henri-Bourassa, Québec, QC G1J 5B3, Canada; 3Department of Radiology, Radio-Oncology and Nuclear Medicine, UdeM, Pavillon Roger-Gaudry S-716, 2900 Boul. Édouard Montpetit, Montréal, QC H3C 3J7, Canada

**Keywords:** [^13^N]Ammonia, cGMP production, semi-automated production system, PET imaging

## Abstract

[^13^N]Ammonia is one of the most commonly used Positron Emission Tomography (PET) radiotracers in humans to assess myocardial perfusion and measure myocardial blood flow. Here, we report a reliable semi-automated process to manufacture large quantities of [^13^N]ammonia in high purity by proton-irradiation of a 10 mM aqueous ethanol solution using an in-target process under aseptic conditions. Our simplified production system is based on two syringe driver units and an in-line anion-exchange purification for up to three consecutive productions of ~30 GBq (~800 mCi) (radiochemical yield = 69 ± 3% n.d.c) per day. The total manufacturing time, including purification, sterile filtration, reformulation, and quality control (QC) analyses performed before batch release, is approximately 11 min from the End of Bombardment (EOB). The drug product complies with FDA/USP specifications and is supplied in a multidose vial allowing for two doses per patient, two patients per batch (4 doses/batch) on two separate PET scanners simultaneously. After four years of use, this production system has proved to be easy to operate and maintain at low costs. Over the last four years, more than 1000 patients have been imaged using this simplified procedure, demonstrating its reliability for the routine production of large quantities of current Good Manufacturing Practices (cGMP)-compliant [^13^N]ammonia for human use.

## 1. Introduction

[^13^N]Ammonia is one of the most commonly used Positron Emission Tomography (PET) radiotracers to perform myocardial perfusion imaging and to quantitatively measure myocardial blood flow and myocardial flow reserve in patients with suspected coronary artery disease and multiple other cardiac conditions [1,2]. The [^13^N]ammonium cation is extracted from the blood and metabolically trapped in the tissue of interest, including myocardium, mainly as [^13^N]glutamine according to blood flow [3]. While [^201^Tl]thallium chloride, [^99m^Tc]technetium sestamibi and [^99m^Tc]technetium tetrofosmin provide qualitative images of relative blood flow distribution, [^13^N]ammonium cation allows both qualitative measurements of relative flow and myocardial blood flow quantification [4,5]. Unlike Single Photon Emission Computed Tomography (SPECT) radioisotopes (i.e., thallium-201 and technetium-99m), the positron-emitting radionuclide nitrogen-13 has two 511 keV annihilation gamma rays detected in coincidence by the PET scanner to accurately assess the blood-flow dependent distribution of [^13^N]ammonia in the heart and quantitatively measure flow [6].

[^82^Rb]RbCl and [^15^O]water are also used for the evaluation of myocardial blood flow as positron-emitting radiopharmaceuticals. However, to date, only [^13^N]ammonia and generator-produced [^82^Rb]RbCl are FDA approved as PET myocardial perfusion agents. While a rubidium-82 generator is certainly an interesting option for centers with a high volume of patients, [^13^N]ammonia is an attractive option for centers with an on-site cyclotron, as it exhibits better physical characteristics for PET imaging [7], such as lower positron energy of N-13 1.199 MeV (β^+^ E_max_) versus Rb-82 3.378 MeV (β^+^ E_max_) reducing the positron range significantly before annihilation with an electron and thus improving image resolution [8]. However, the nitrogen-13 half-life of 9.96 min makes the manufacturing and time delivery of [^13^N]ammonia challenging. Hence, different strategies for [^13^N]NH_3_ production have been adopted by several labs varying from simple in-target production (without further purification) to more complex procedures with in-line purification, dedicated production systems or automated synthesis modules. The in-target production of [^13^N]ammonia with aqueous ethanol (5 mM) as a radical scavenger under pressurized conditions is considered the classic synthetic route [9]. Nonetheless, the absence of purification steps renders the final formulation less than optimal due to the low isotonicity and potential presence of radionuclidic contaminants from the target material. To circumvent these problems, a purification process through an anion-exchange resin has been utilized to remove all anionic impurities with a subsequent reformulation in saline for dispensing ^13^[N]NH_3_ in the form of ^13^[N]NH_4_^+^ ions [10].

Complex dedicated production systems have been evaluated to increase product yields and faster manufacturing processes. Frank et al. [11] proposed a dedicated production system consisting of nine parallel Waters CM-Light Sep-Pak cartridges connected to an ISAR chip. The compact microfluidic system allows nine consecutive productions to produce [^13^N]ammonia at >96% radiochemical yield (RCY) per batch, including approximately five minutes for the purification and reformulation processes. Despite all these advantages, this system is yet in the prototype stage and still needs to be thoroughly evaluated. In recent years, the widespread use of commercial synthesis modules was also applied in the automated production of ^13^[N]ammonia. Several examples can be found in the literature, with a synthesis time of around five minutes and RCYs of 90% or higher. Notably, Yokell et al. [12] set out a simple method that could also be adapted and validated on commercially available synthesis units. This approach utilizes quaternary methyl ammonium (QMA) and carboxymethyl (CM) cartridges in series to accomplish the purification, formulation, and sterile filtration of [^13^N]ammonia (21 ± 2 GBq, 572 ± 60 mCi) in about five minutes. Considering that [^13^N]NH_3_ is among the oldest clinically used PET radiopharmaceuticals whose cost have historically been low (due to the low cost of the target material and the process), the price and availability in radiochemistry laboratories of such commercial modules and their often-cumbersome usage to customized productions have limited its widespread utilization in nuclear medicine.

In this paper, we present our institutional experience to reliably produce [^13^N]ammonia in large quantities for clinical use. The proposed [^13^N]ammonia production system has been in operation since 2018 allowing an average of two doses per patient, two patients per batch (four doses/batch) on two separate PET scanners simultaneously. So far, more than 1000 patients have been scanned, demonstrating system reliability for the routine production of large quantities of [^13^N]ammonia for human use.

## 2. Results

The results of six productions (two consecutive batches per day) using the 2XL target at 10 mM aqueous ethanol solution are presented in Table 1. All batches were found to meet the FDA/USP product specifications with slight variations of pH but within the acceptance range (4.5–7.5 at time of injection) after extensive dilution of [^13^N]NH_3_ with 0.9% sodium chloride in the nuclear medicine PET unit before administration to patients. Analytical High-Performance Liquid Chromatography (HPLC) chromatograms for [^nat^N]NH_4_Cl standard and [^13^N]ammonia samples are presented in Figure 1a–c. The radiochemical identity was determined by peak comparison of the retention times (t_R_) between the conductivity peaks of [^nat^N]ammonium chloride and [^13^N]ammonia (both at t_R_ = 2.83 min, Figure 1a,b). The radiation signal from [^13^N]ammonia came out after the conductivity peak (t_R_ = 3.17 min, Figure 1c) since the radiation detector is placed after the conductivity detector. In this HPLC system, the [^13^N]ammonium cation can be adequately separated from the known negatively charged impurities, [^18^F]fluoride and [^13^N]nitrous oxide (NO_x_^-^) eluted with the void volume of IC-Pak cation exchange column. None of these impurities were observed in any [^13^N]ammonia batches. Particular interest was given to the analysis of long-lived radionuclidic impurities by HP(Ge) spectrometry, where no detectable radionuclide contaminants from the cyclotron target body or window were observed in any validation runs. The system described in this article was able to routinely produce 29 ± 2 GBq (793 ± 49 mCi, n = 345) of [^13^N]ammonia per batch with the cyclotron 2XL target at 55 µA for up to 40 min of bombardment. [^13^N]Ammonia was produced in approximately five minutes at yields higher than 90% (d.c.). The whole process, including QC testing until batch release, was carried out in approximately 11 min from EOB. The product was found to be stable at room temperature up to 60 min after EOM, and thus, expiry time was set at 60 min post-EOM. To meet clinical needs, up to three batches can be produced per day.

## 3. Discussion

Manufacturing large quantities of [^13^N]ammonia for patient PET studies can be challenging, taking into account the duration of the methodological process and the quality control (QC) assays. For instance, producing [^13^N]NH_3_ using Devarda’s alloy process usually takes 20 min. However, the determination of residual alumina is mandatory, adding an extra QC test before release [13,14]. Despite the fact that the final formulation usually displays values within specifications, this approach has an inevitable drawback given the length of the whole process before batch release and the short half-life of nitrogen-13 (9.96 min) preventing access to large quantities of [^13^N]ammonia for patients. The introduction of ethanol (5–10 mM in water) as a radical scavenger in the target solution has become the current method of choice for the in-target production of [^13^N]ammonia. In this manuscript, manufacturing of large quantities of [^13^N]ammonia in high purity was carried out by proton-irradiation of a 10 mM aqueous ethanol solution followed by QMA cartridge purification and sterile filtration using a dedicated semi-automatic system designed to speed up and simplify the production process. In addition to the simplicity of the method, the fact that it does not require an automated radiosynthesis unit or expensive cassette-based systems sold in sealed sterile containers reduces the costs of the whole process, which adds to the workflow of operations.

The production system comprises two syringe drivers, both connected to the cyclotron 2XL target and the cyclotron helium-unload mechanism to transfer liquids to a class A dispensing hot cell. The rinsing syringe driver was programmed to work separately from the cyclotron software using a minimalist syntax in MS-DOS language. Two dedicated homemade short programs for rinsing and cleaning the cyclotron transfer lines from the target to the dispensing hot cell are carried out by a simple “one-click” procedure. These processes were intended to facilitate the chemist’s work during the preparation of the production and for the post-production sanitization with ethanol to achieve manufacturing with current Good Manufacturing Practices (cGMP) compliance. Before the first batch of the day, the target and transfer lines are rinsed with the freshly prepared target solution, followed by the preconditioning and drying of the QMA Sep-Pak cartridge. The design of a serial arrangement of a QMA cartridge and a 0.22 µm filter allowing purification and sterile filtration in-series has a considerable impact on shortening the duration of the production process and on the reduction in manipulation errors. In fact, each batch takes approximately five minutes for completion. Considering the presence of radionuclide impurities that come from the activation of HAVAR foils and niobium material and the natural presence of 0.2% oxygen-18 in target water (that produces fluorine-18) [15,16,17], most [^13^N]ammonia production processes use in-series anion-cation cartridges aided by valves to purify [^13^N]NH_3_ [11,12,18]. Hence, an anion exchange resin is mainly utilized to trap negatively charged impurities ([^18^F]F^−^, [^13^N]NO_x_^−^ or [^48^V]VO_4_^3−^), and a cation exchange resin is used to reformulate [^13^N]NH_4_^+^ in 0.9% saline [14]. According to our experience, the fact that our system does not utilize the trap and release of [^13^N]NH_4_^+^ (available in most automated synthesis modules) can be considered an advantage since operators do not need to turn valves with manipulator arms to purify and isolate [^13^N]ammonia for reformulation. However, our approach requires sufficient dilution with saline to guarantee the pH and isotonicity of the final [^13^N]ammonia formulation.

Considering the aforementioned, one might think that a set of two ion exchange (IEX) cartridges is strictly necessary to have the radionuclidic contaminants at their lowest level during [^13^N]ammonia production. However, several authors have demonstrated that using only a CM cartridge (cation exchange) to selectively trap [^13^N]NH_4_^+^ from anionic impurities in the flow-through is sufficient to achieve more than 99.96% radionuclidic purity [10,11,19,20]. Taking this into account, we based our simplified procedure on a single QMA cartridge which gave [^13^N]ammonia with a radionuclidic purity higher than 99.5%. As reported by Bormans et al. [21], the purification of [^13^N]NH_3_ with an anion exchange cartridge (5 × 10 mm, Dowex AG 1-X8) also revealed the presence of other unidentified radionuclidic impurities but in low concentration (≤0.001% EOB) after two days decay analysis by gamma spectrometry.

Given the duration of PET scans usually performed at rest/stress, batches of around 26–33 GBq (717–885 mCi) are normally produced back-to-back to fulfill patient demands in our PET unit. The production time and yields were similar to those using other prototype systems [11,18,19] or commercial synthesis modules [12,20]. Therefore, our simplified manufacturing method allows the production of large quantities of [^13^N]ammonia with less automation or manual interventions, thus providing an alternative to laboratories without access to automated technology. To date, the number of lots produced since 2018 demonstrates the robustness and reliability of our simplified cGMP-compliant [^13^N]NH_3_ production system for human use. Importantly, with this production set-up, we continue to guarantee consistent and uninterrupted on-demand [^13^N]ammonia to meet clinical needs.

## 4. Materials and Methods

### 4.1. Chemical and Reagents

All chemicals and reagents were commercially available and used without further purification.

### 4.2. System Description

Materials:

The system consists of two syringe pumps (model: V6 48K, Norgren, Denver, CO, USA) installed in a panel located outside the vault, a six-position Valco valve (model: C5-2036, VICI, Houston, TX, USA) and the nitrogen-13 target controlled by the IBA C18/9 cyclotron (software version: Bill), a four ports valve (model: C2-3184UMH, VICI, Houston, TX, USA) that allows selecting between helium or target solution, and a pneumatic Valco valve (model: C5-2034UMH, VICI, Houston, TX, USA) to select the liquid transfer between two dispensing hot cells (optional).
System operation:

The syringe pump is controlled by volume, valve position, number of cycles, and speed with a dedicated laptop located in the production cleanroom (ISO 7/class 10,000). A programmed script file (Batch file) sends the signal to the syringe pump board to perform the required tasks during the production process. The operator selects the execution of specific files depending on where the [^13^N]ammonia manufacturing is. For rinsing the target and transfer lines (Figure 2(a1)), the operator sets the cyclotron valve position to ”Unload” to connect the target solution and the transfer line. Then, the corresponding script file is selected, and the pump starts pushing the target solution to fill all lines and the cyclotron target. To rinse the Overflow line (Figure 2(a2)), the operator sets the cyclotron Target valve on the “Load” position and starts several successive fillings to rinse this line. The target overflow goes directly to a bottle installed beside the cyclotron. At the end of these operations, the lines are dried with high-purity helium (grade 5.0) for a few minutes by activating the three-way valve. The target is then loaded with the target solution using the loading system before starting the beam (Figure 2b). Once the target is loaded, the cyclotron valve software closes all ports, and the system is ready to start the beam. After beaming, the activity is transferred with He from the target to the selected dispensing hot cell (Figure 2c) by changing the position of the cyclotron Target valve to unload.

### 4.3. Production of [^13^N]Ammonia Injection

[^13^N]Ammonia is manufactured in two steps, starting with the nitrogen-13 production in the cyclotron target followed by the reformulation and filtration to render [^13^N]ammonia isotonic, sterile and non-pyrogenic. Prior to each production, the transfer line from the cyclotron to the class A dispensing hot cell (Comecer, Castel Bolognese, Italy, ISO 5/class 100) is rinsed with a freshly prepared 60 mL target solution (10 mM ethanol USP in water for injection (WFI) using the rinsing system presented in Figure 2(a1). Then, the anion exchange cartridge (Sep-Pak Accell plus^®^ QMA plus light, Waters, Milford, MA, USA) is connected to the cyclotron line as depicted in Figure 3a and eluted with 50 mL target solution for eight minutes (Figure 2(a1)), followed by drying with He for five minutes. Each month, the production system (target and delivery line) is sanitized with ethanol USP followed by five flushes of 10 mL target solution.
[^13^N]Ammonia is produced with a Cyclone^®^ 18/9 cyclotron (IBA, Louvain-La-Neuve, Belgium) via the nuclear reaction ^16^O(p,α)^13^N with 18-MeV protons degraded to 16 MeV using an aluminum degrader (300–400 µm) installed in the target collimator to prevent the production of oxygen-15 via the ^16^O(p,pn)^15^O nuclear reaction. The 2XL target is filled with 3.5 mL of target solution (10 mM ethanol in WFI) (Figure 2(a2)) and bombarded at a current of 55 µA for 20–40 min (Figure 2b). The target body is made of niobium, and the entrance windows are made of a 12.5 µm thick titanium foil (ø 23.5 mm) and a 35 µm thick Havar foil (ø 43 mm).The set-up of the Class A dispensing hot cell consists of two vial assemblies containing either four (first batch of the day) or three (second batch) 10 mL vials (Hollister Stier, Montréal, QC, Canada), enough syringes and needles of the appropriate size for the manipulation to be performed, along with a 3 mL syringe containing 2.5 mL of 3% saline. Using the manipulating gloves, before the transfer of the dose, a hydrophilic 0.22 μm vented filter (Millex GV, Millipore, Burlington, MA, USA) and a venting 0.22 μm filter (PharmAssure, Pall, Port Washington, NY, USA) are inserted into the patient (receiving) 10 mL vial. The vented filter is then connected to the rinsed and dried (see above) QMA + delivery line, as shown in Figure 3a. The remaining three vials are used for QC, retain and sterility/endotoxin samples, respectively (Figure 3b). Since the sterility sample of the second batch is pooled with the first one, the vial assembly for the second batch contains only three 10 mL vials. An acrylic plate with three pH strips (0 to 6) and a vented 100 mL vial full of target solution to perform the filter integrity test complete the setup for the [^13^N]ammonia production.At the EOB, the irradiated water/ethanol/[^13^N]NH_3_ solution is flushed out of the target (Figure 2c) with 0.25–0.5 MPa (2.5–5 bar) flow of He through an in-line QMA cartridge to trap any anionic impurities such as [^18^F]fluoride and [^13^N]NO_x_^−^, into a dispensing hot cell previously set up for the aseptic preparation (see above). Right after, an additional target rinse with 2.5 mL of target solution is performed to recover the remaining activity.Approximately 0.1 mL of [^13^N]NH_3_ is withdrawn to start the QC, then 2.5 mL of 3% saline is added to the bulk vial to produce isotonic [^13^N]ammonia final product. From this, it is understood that the QC steps start before finishing the sampling and reformulation of the final product (both stages overlapping). Sterility/endotoxin and retain samples (0.3 mL sterility/endotoxin + 0.1 mL retain) are taken at this step of the process (Figure 3b) while visual inspection, pH and filter integrity tests are performed subsequently. Sterility/endotoxin samples of all batches of the day are pooled, while retained samples from each batch are kept separately in case a sterility and/or endotoxin re-test is required afterwards.The final [^13^N]ammonia product is removed from the dispensing hot cell using a dedicated drawer (Figure 3b) and transferred to the Nuclear Medicine department in less than a minute by a built-in pneumatic system while the QC assays are performed in order to save time.

After EOB, the total manufacturing time from the target, purification (via QMA), sterile filtration, reformulation and QC testing until the batch release of [^13^N]ammonia is approximately 11 min. Our validation studies have revealed that the same steps as described above can be repeated for up to three cGMP-compliant back-to-back production batches per day. In that case, the same target solution and QMA are used, but different vial assemblies containing new product vials, filters and syringes are changed after each batch of the day.

### 4.4. Quality Control Testing of [^13^N]Ammonia Injection

The quality control tests of [^13^N ammonia are performed for each batch in about eight minutes (some QC assays are started before the EOM in order to save time) to ensure the final drug product meets all the specifications for human use. Radiochemical identity and purity are determined by analytical HPLC (column: Waters IC-Pak Cation exchange M/D 150 × 3.9 mm, mobile phase: 3 mM HNO_3_/0.1 mM EDTA, flow rate: 1.3 mL/min) using a conductivity (Waters 432, Milford, MA, USA) and radioactivity (Raytest Gabi Star, Straubenhardt, Germany) detectors. The retention time of [^13^N]NH_3_ is compared to that of the [^nat^N]NH_4_Cl reference standard solution and must be ± 10% (see example in Figure 1). Stability testing by analytical HPLC was conducted at 0, 30, and 60 min after EOM. Radionuclidic identity is assayed for half-life (10 ± 1 min) and annihilation peak (466–556 keV) using a Capintec CRC^®^-15tW Dose Calibrator containing a NaI(Tl) well-counter coupled to a multichannel analyzer. Radionuclidic purity (>99.5%) is analyzed by HP(Ge) gamma spectrometry as a periodic quality indicator test during the validation runs and re-tested annually afterwards at Nuclear Activation Analysis Lab (Ecole Polytechnique, Montréal, QC, Canada). Bacterial endotoxins (<175 EU per dose) are quantified as a post-release test by LAL EndoSafe Portable Testing System (Charles River, Wilmington, MA, USA) on the pooled samples from each batch of the day. Sterility testing is assayed (on the pooled samples) for no growth after 14 days following USP 71 with a contracted organization (Isologic Innovative Radiopharmaceuticals, Montréal, QC, Canada).

## 5. Conclusions

Our institutional experience of producing large quantities of [^13^N]ammonia using two syringe drivers and in-line anion-exchange resin purification have been thoroughly validated after four years of continuous production. Hence, [^13^N]NH_3_ injection was found to meet all the product specifications. The simplified production system described in this work has been demonstrated to be reliable and easy to operate and maintain, aiding to keep the [^13^N]ammonia production costs low while satisfying clinical needs.

## Figures and Tables

**Figure 1 molecules-28-04517-f001:**
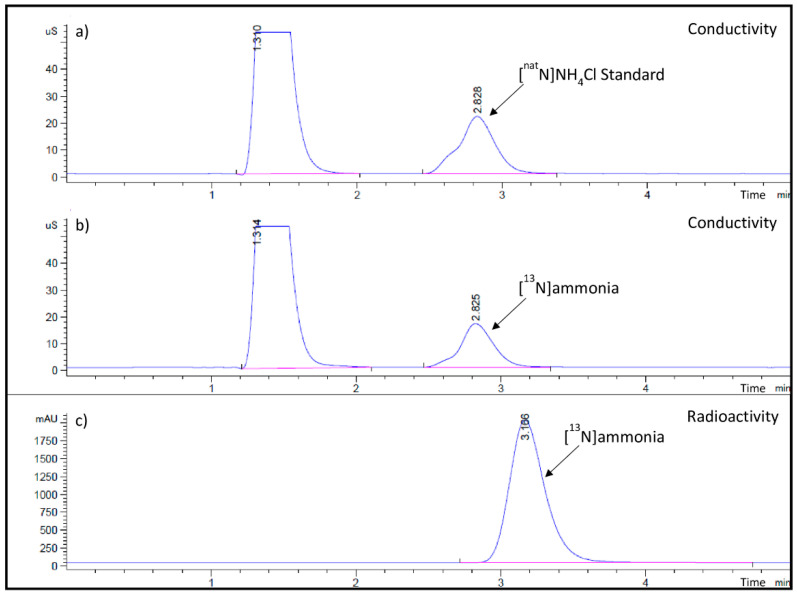
Representative analytical HPLC chromatograms for [^13^N]ammonia production: (**a**) [^nat^N]ammonium chloride standard by conductivity detection, t_R_ = 2.83 min; (**b**) [^13^N]ammonia sample analysis from product vial by conductivity detection, t_R_ = 2.83 min; (**c**) [^13^N]ammonia sample analysis from product vial by radioactivity detection, t_R_ = 3.17 min. Differences in retention times are intrinsic to the HPLC system due to the fact that the radiation detector is placed after the conductivity detector.

**Figure 2 molecules-28-04517-f002:**
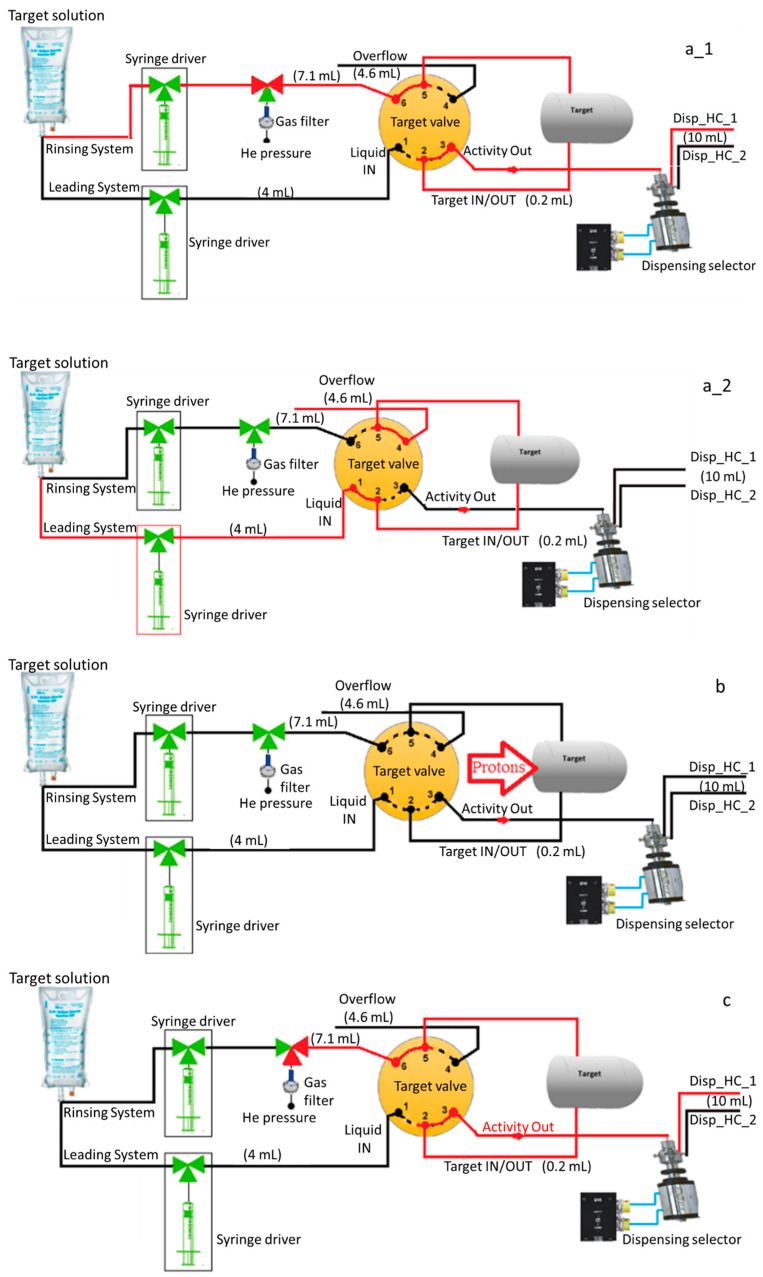
Schemes describing the [^13^N]ammonia production system consisting of two syringe drivers connected to the cyclotron target. The line highlighted in red displays the production workflow: (**a1**) For rinsing, a dedicated syringe driver transfers the freshly prepared target solution through the target towards the class A dispensing hot cell (Disp_HC); (**a2**) For loading, a separate syringe driver fills the target prior to the bombardment; (**b**) Beaming sequence; and (**c**) Unloading sequence with helium and transfer to the selected dispensing hot cell.

**Figure 3 molecules-28-04517-f003:**
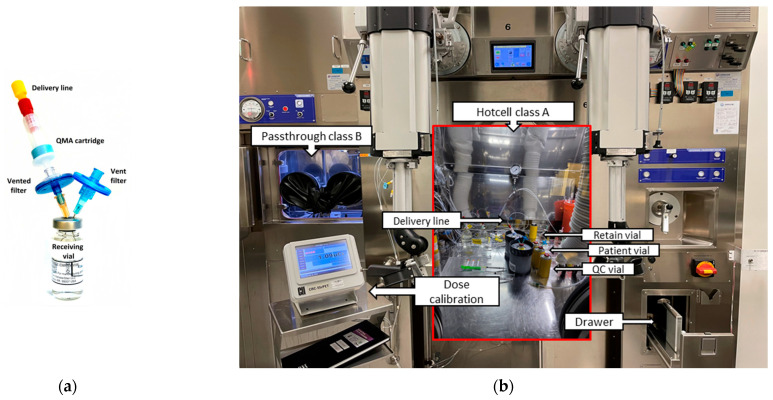
Aseptic set-up for [^13^N]ammonia production: (**a**) Receiving formulation vial and connection from cyclotron target; (**b**) Class A dispensing hot cell with a class B pre-chamber (passthrough) for material entrance.

**Table 1 molecules-28-04517-t001:** Quality control test results of [^13^N]ammonia for three production days (two consecutive batches per day).

Specification	Acceptance Criteria	Day 1 *	Day 2 *	Day 3 *
Activity EOB GBq/mCi	-	43.03/1163 44.55/1204	40.33/1090 42.18/1140	42.74/1155 43.07/1164
Activity EOM (GBq/mCi)	-	28.19/762 28.82/779	29.23/790 29.97/810	29.49/797 29.49/797
Elapsed time EOB to EOM	-	5 min 5 min, 15 s	4 min, 40 s 4 min, 42 s	4 min, 55 s 5 min, 15 s
Yield (d.c. to EOB)	-	0.93 0.93	1.00 0.99	0.97 0.99
Appearance	Clear colourless solution with no particulate matter	Pass	Pass	Pass
pH	3.5–8.5	4.0 4.0	4.7 4.4	4.5 4.0
Filter integrity	≥manufacturer specification of 0.34 MPa (50 psi)	Pass	Pass	Pass
Radionuclide identity	Half-life: t_1/2_ = 10 ± 1 min	9.03 min 9.84 min	9.86 min 9.91 min	10.39 min 9.95 min
	Photopeak: 466 keV < E < 556 keV	Pass	Pass	Pass
Radiochemical identity and Purity	HPLC t_R_ 3.0 to 3.8 min	3.17 min 3.27 min	3.33 min 3.25 min	3.26 min 3.19 min
	Purity ≥ 95%	99.99% 99.98%	99.99% 99.99%	99.98% 99.99%
Radionuclidic purity	Purity ≥ 99.5%	99.98% 99.92%	99.99% 99.99%	99.74% 99.98%
Bacterial endotoxins ^#^	≤175 EU/vial	Pass	Pass	Pass
Sterility ^#^	No growth after 14 days	Pass	Pass	Pass

* Second lines correspond to the data from the second batch of the day. ^#^ Test performed with the pooled fraction of all batches of the day. Abbreviations: EOB: End of Bombardment, EOM: End of Manipulation, d.c.: decay corrected, t_R_: retention time, EU: endotoxin units. Note: “Pass” means all six productions passed QC.

## Data Availability

Not applicable.

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
