# Peer review of "A Reliable Production System of Large Quantities of [13N]Ammonia for Multiple Human Injections"

_molecules, 2023, doi:10.3390/molecules28114517_

Round 1

Reviewer 1 Report

Simple productions of short-lived radiotracers is vital to keep their costs low and to meet clinical demand. The authors are commended for their simple system but their publication has not adequately informed the reader how to assemble said device. The details to produce and assess the quality of [13N]ammonia are not novel nor different from other sites. The most important feature of this report is the semi-automated device. No details on the components used to assemble or software used to control this device were given. In a supplemental file, please provide details of components used, wiring, and programming such that other sites may replicate and use this technology.

There were a few major surprises from this method.

1) that there were no long lived isotopes produced by using sterile water for injection in the target solution rather than ultrapure water.  What was the metal content of the WFI used in making target solution? 

2) Ethanol is used (10 mM is 10% of the PDE for ethanol) in the target and to clean delivery lines, can the authors justify why no residual testing was completed prior to release? While the concentration will be low, deviations from normal could be trended to point out a failing system before a complete failure. 

3) It was also strange that the pH value failed but was deemed acceptable when diluted in the nuclear medicine lab. This is atypical and would be a non-compliance from regulators. This could have been solved by using an acceptable buffer for injections (PBS). Can the authors explain how they justified releasing material that was out of specification for patient injection? Is there validation data to ensure that dilution is sufficient to bring the product into specification?

4) Likewise, is there validation for the bacteria load of the delivery lines from the cyclotron to the grade A hotcell? They are cleaned monthly; is that enough? The authors state their method using one cartridge is sufficient to purify the [13N]ammonia but the target solution is getting added directly to the final patient vials. While slightly more effort, the catch-and-release approaches using a cationic exchange cartridge removes this solution and any bacterial (and endotoxins) from the target delivery lines into a waste vial prior to reformulation into a patient dose vial. This extra step also seems to help the low pH. Although no endotoxins or bacteria were seen in 4 years of use, limiting bioburden as much as possible is always a goal for cGMP. 

5) QC was not conducted on the final formulated product. Saline was added after the QC sample was taken. This is odd and seems to contravene GMP. Is that a typographical error? 

In the discussion section, the produced activity was listed as 26 - 33 Gbq. Is that at end of bombardment or end of purification? What is the stability of those batches, that is, what is the expiry time? Has shelf-life testing been completed? 

It is unclear if the same filter and QMA are used for multiple batches of ammonia. The conditioning and drying are completed using the device but it is not stated whether a new vial/QMA/filter set-up is used for each production. Please clarify.

Reviewer 2 Report

see reviewed manuscript with corrections, suggestions and questions

Reviewer 3 Report

In the paper entitled “A Reliable Production System of Large Quantities of [13N]Ammonia for Multiple Human Injections” the authors present an overview of experimental work underlying the production and synthesis of [13N]Ammonia for PET imaging.

Results are described in a very superficial way, and I believe that more evidence about acquired experience in several runs of the production process could be shared. Results should be presented in tables that demonstrate variables achieved, such as: EOB produced activity per run, radiochemical yield, pH, radionuclidic purity, biological quality of the final product, etc.

The discussion do not reflect the level of novelty of this work, since it does not compare results obtained with other published reports nor highlights different features of the procedure undertaken compared to other processes already on place on other centers.

Thus, I believe that this paper is acceptable but should beneficiate from a minor review in order to improve its scientific value.

Round 2

Reviewer 1 Report

Thank you for the added details and thorough responses to the previous comments. The design specification are clear and easy to follow should someone wish to implement such a system in their own facility.

To clarify on the ethanol - if not part of the final formulation, the daily limit is 5000 ppm, not 10% v/v. Thus the target solution (albeit undiluted) is about 10% of that daily limit. Regardless, the justification for limited GC testing is sufficient.

For the other point (bioburden and pH), it's wonderful to see some leeway from Health Canada on something with such a short half-life. I disagree that adding the catch and release, two cartridge method would add a detrimental amount of time and it would further de-risk many potential contaminants (long-lived isotopes, bioburden, ethanol). That said, those risks are already very low as eloquently explained in your cover letter. After reviewing this paper, I now wish to rethink our production workflow to evaluate if we are over engineering controls that are burdening our production!

Reviewer 2 Report

The quality of the manuscript has been significantly improved and it can now be accepted for publication. I have only a few editorial comments:
- Table 1: the authors use several times the abbreviation seg in the time indication (e.g. 4 min, 40 seg). I suppose this has to be sec instead of seg, and correctly, it should then be s instead of sec as official abbreviation. Maybe it has to be checked with the authors whether they mean seconds (I have no idea what seg could be the abbreviation for)
- page 5: ...contaminants at its lowest level ... should read ...contaminants at their lowest level ...
- page 8: the abbreviation WFI is used without explanation. Although pharmacists will know that WFI stands for water for injection, not all readers (including radiochemists) will know this. It would be better to write this in full.
So, my recommendation is 'accept for publication', with a few editorial corrections.
